# Prognosis of Pneumonia in Head and Neck Squamous Cell Carcinoma Patients Who Received Concurrent Chemoradiotherapy

**DOI:** 10.3390/biomedicines12071480

**Published:** 2024-07-04

**Authors:** Tzu-Hsun Kao, Tai-Jan Chiu, Ching-Nung Wu, Shao-Chun Wu, Wei-Chih Chen, Yao-Hsu Yang, Yu-Ming Wang, Sheng-Dean Luo

**Affiliations:** 1Department of Hematology-Oncology, Kaohsiung Chang Gung Memorial Hospital, Chang Gung University College of Medicine, Kaohsiung 833, Taiwan; ely3vmp4@gmail.com (T.-H.K.); kuerten@cgmh.org.tw (T.-J.C.); 2Department of Otolaryngology, Kaohsiung Chang Gung Memorial Hospital, Chang Gung University College of Medicine, Kaohsiung 833, Taiwan; taytay@cgmh.org.tw (C.-N.W.); jarva@cgmh.org.tw (W.-C.C.); 3Department of Anesthesiology, Kaohsiung Chang Gung Memorial Hospital, Chang Gung University College of Medicine, Kaohsiung 833, Taiwan; shaochunwu@gmail.com; 4Graduate Institute of Clinical Medical Sciences, College of Medicine, Chang Gung University, Taoyuan 333, Taiwan; 5Department of Traditional Chinese Medicine, Chang Gung Memorial Hospital, Chiayi 613, Taiwan; r95841012@cgmh.org.tw; 6Health Information and Epidemiology Laboratory of Chang Gung Memorial Hospital, Chiayi 613, Taiwan; 7School of Traditional Chinese Medicine, College of Medicine, Chang Gung University, Taoyuan 333, Taiwan; scorpion@cgmh.org.tw; 8Department of Radiation Oncology, Kaohsiung Chang Gung Memorial Hospital, Chang Gung University College of Medicine, Kaohsiung 833, Taiwan; 9School of Medicine, College of Medicine, National Sun Yat-sen University, Kaohsiung 804, Taiwan

**Keywords:** head and neck cancer, survival, chemoradiation, pneumonia

## Abstract

Concurrent chemoradiotherapy (CCRT) is the standard treatment for patients with locally advanced squamous cell carcinoma of the head and neck (HNSCC). Pneumonia is a significant complication in these patients. This study aims to identify pneumonia risk factors and their impact on survival in HNSCC patients undergoing CCRT. Data from the Chang Gung Research Database (CGRD) were retrospectively reviewed for patients treated between January 2007 and December 2019. Of 6959 patients, 1601 (23.01%) developed pneumonia, resulting in a median overall survival (OS) of 1.2 years compared to 4.9 years in the non-pneumonia group (*p* < 0.001). The pneumonia group included older patients with advanced tumors, more patients with diabetes mellitus (DM), more patients with invasive procedures, longer chemotherapy and radiotherapy durations, and lower body weight. The 2-year, 5-year, and 10-year OS rates were significantly lower in the pneumonia group. Multivariate analysis identified alcohol consumption, DM, gastrostomy, nasogastric tube use, longer chemotherapy, and a 2-week radiotherapy delay as independent risk factors. Understanding these risks can lead to early interventions to prevent severe pneumonia-related complications. A better understanding of the risks of pneumonia enables early and aggressive interventions to prevent severe complications.

## 1. Introduction

Squamous cell carcinoma of the head and neck (HNSCC) arises from the mucosal surfaces lining various regions including the oral and sinus cavities, oropharynx, hypopharynx, and larynx. HNSCC ranks as the seventh most prevalent cancer type worldwide [1], but within Taiwan, it holds the position as the sixth most commonly diagnosed cancer and the fifth most fatal. Notably, HNSCC exhibits a higher incidence among men compared to women. Specifically in Taiwan, males rank third in terms of HNSCC incidence and fourth in terms of mortality rate associated with the disease [2].

The majority of patients diagnosed with HNSCC typically present with locally advanced disease, necessitating a multifaceted and aggressive treatment approach aimed at achieving a cure while minimizing treatment-related toxicity. Complete tumor resection remains the cornerstone of treatment for localized HNSCC. Concurrent chemoradiotherapy (CCRT) is the standard treatment for HNSCC when surgical resection is not feasible or would compromise long-term functional outcomes [3,4]. In adjuvant therapy following surgery for HNSCC, CCRT has been linked with a decreased rate of loco-regional failure and a 6.5% reduction in 5-year mortality compared to radiotherapy alone [5,6].

Patients with HNSCC undergoing CCRT commonly encounter adverse effects such as profound oral inflammation, excruciating discomfort, persistent queasiness, irreversible dryness of the mouth, and difficulties in swallowing. Swallowing dysfunction often correlates with malnutrition, aspiration pneumonia, and obscured scheduled treatment [7,8,9]. In many HNSCC patients, such toxicities are severe and life-threatening. The documented occurrence of aspiration pneumonia varies within the range of 5 to 25.4% among patients with HNSCC undergoing CCRT [10,11,12], and the median time from initiating radiotherapy to an aspiration pneumonia event is 5 months in HNSCC patients [10]. To prevent choking and suffocation, HNSCC patients were often suggested to administer enteral nutrition with a nasogastric tube and gastrostomy. Nevertheless, the use of nasogastric tubes often leads to persistent discomfort in the nasal and pharyngeal regions for patients, while gastrostomy carries an elevated risk of wound infection and sepsis. Moreover, patients deemed at high risk of airway obstruction were advised to undergo tracheostomy as a measure to safeguard the airway. While there have been limited studies exploring predictive and prognostic factors for pneumonia in HNSCC patients undergoing CCRT, the precise characteristics remain unclear. Shirasu et al. reported that suboptimal oral hygiene, advanced nodal classification, pre-treatment hypoalbuminemia, and hospitalization for treatment were identified as independent risk factors in HNSCC patients undergoing CCRT or bio-radiotherapy [12]. Patil et al. just found that pre-treatment dysphagia was correlated with aspiration pneumonia [13]. Therefore, we aimed to investigate the survival condition and related pneumonia risk factors for HNSCC patients with CCRT.

## 2. Materials and Methods

### 2.1. Patient Recruitment

This retrospective study received approval from the Institutional Review Board (IRB) of Kaohsiung Chang Gung Memorial Hospital, with the protocol number 202101065B0. In accordance with these protocols and IRB regulations, informed consent was waived due to the study’s design. All methods were conducted in compliance with pertinent guidelines and regulations.

### 2.2. Study Design and Subjects

Using the Chang Gung Research Database (CGRD), we searched non-metastatic head and neck cancer patients who underwent CCRT at four Chang Gung Memorial Hospitals in Taiwan. Head and neck cancer was diagnosed with typical findings from computed tomography (CT) or magnetic resonance imaging (MRI) and pathological reports, and was identified based on the International Classification of Diseases, 9th and 10th Revision (ICD-9/ICD-10) diagnosis code in the CGRD database, as provided in Appendix A (including C00-C13, C32; excluding C07, C08, C11). Patients should undergo an ultrasound of the abdomen, a bone scan, or positron emission tomography to ensure comprehensive tumor staging. The inclusion criteria were defined as follows: (1) age greater than 20 years old, (2) pathologically confirmed squamous cell carcinoma, and (3) received CCRT. The exclusion criteria included the following: (1) diagnosis of esophageal cancer or Salivary gland cancer before diagnosis of HNSCC (N = 198), (2) distant metastases (N = 379), (3) data of tumor staging missing (N = 14), and (4) coding of pneumonia within 90 days before diagnosis of HNSCC (N = 40). Nasopharyngeal carcinoma (NPC) was also excluded from this study due to its different pathology, clinical course, and treatment (N = 2139). CCRT consisted of adjuvant and definitive treatments in this retrospective study.

Concurrent chemotherapy was defined as the administration of chemotherapy from 2 weeks before the initiation of radiation therapy to 2 weeks after completing radiation. The chemotherapy regimen consisted of either weekly cisplatin at doses ranging from 30 to 40 mg/m^2^ or two cycles of carboplatin with an area under the curve (AUC) of 5. Radiotherapy techniques included conventional three-dimensional conformal radiation therapy (3D-CRT) or intensity-modulated radiation therapy (IMRT). The accumulated radiation doses were 70–76 Gy for the primary tumor, 60–70 Gy for positive lymph node lesions in the neck, and 50 Gy for uninvolved lymphatics in the neck.

Pneumonia, the primary endpoint of this study, was identified from the ICD-9/ICD-10 diagnosis code in the CGRD database, as provided in Appendix A (including J13-J18, J69, J9589; excluding J157, J16). Pneumonia development during CCRT was defined as from diagnosis of HNSCC to within 90 days of finishing CCRT.

The procedures of tracheostomy, gastrostomy, and nasogastric tube were determined after diagnosis of HNSCC within 90 days of finishing CCRT and before diagnosing pneumonia. In total, 6959 HNSCC patients with CCRT were analyzed in our study.

### 2.3. Statistical Analysis

We conducted a comprehensive retrospective analysis of survival data in patients diagnosed with pneumonia. The clinicopathological variables under review included age at diagnosis, gender, primary tumor site, and tumor staging, as classified by the American Joint Committee on Cancer (AJCC) 7th edition. In addition, we collected data on lifestyle habits such as cigarette smoking, alcohol consumption, and betel nut chewing. Medical history factors were also recorded, including the presence of diabetes mellitus (DM), tracheostomy, gastrostomy, nasogastric tube use, esophageal cancer, the duration of chemotherapy and radiotherapy, body weight loss, and any radiotherapy delay of two weeks or more.

For statistical analysis, categorical variables were assessed using Pearson’s chi-square test or Fisher’s exact test, depending on the data distribution. Continuous variables that followed a normal distribution were analyzed with the independent t-test, while non-normally distributed continuous variables were evaluated using the Mann–Whitney U test. This rigorous approach allowed for a detailed examination of the factors influencing survival outcomes in pneumonia patients.

Overall survival (OS) was defined as the duration from the date of diagnosis to death from any cause or the end of the follow-up period. To compare survival rates between different groups, we employed the log-rank test. The Kaplan–Meier method was utilized to generate survival curves, providing a visual representation of the survival distribution. To evaluate the hazard ratios and determine the impact of various factors on survival, we applied the Cox proportional hazards model. This comprehensive statistical approach enabled a robust analysis of survival outcomes in the patient cohort.

Logistic regression was employed to identify the correlation between pneumonia and various clinicopathological factors. After adjusting for potential confounding variables such as gender and tumor stage, we conducted both univariate and multivariate analyses to evaluate risk factors for pneumonia in patients with HNSCC undergoing CCRT. To further minimize bias, we utilized Greedy propensity score matching based on age (±2 years), AJCC stage, and primary tumor site. This method matched 1585 pneumonia patients to 3093 patients without pneumonia in a 1:2 ratio. Univariate and multivariate analyses were then performed to identify risk factors for pneumonia using logistic regression. All statistical tests were two-sided, and a *p*-value of less than 0.05 was considered statistically significant. The analyses were conducted using SAS version 9.4 (SAS Institute, Cary, NC, USA).

## 3. Results

### 3.1. Patient Characteristics

Using the CGRD, we identified 9729 non-metastatic HNSCC patients who underwent CCRT between January 2007 and December 2019. After excluding 631 patients based on exclusion criteria and 2139 patients with NPC, we had 6959 HNSCC patients remaining. Among these, 1601 (23.01%) were identified as having pneumonia, while 5358 (76.99%) did not experience any episodes of pneumonia, as depicted in the flow chart (Figure 1). Table 1 provides a detailed summary of the demographic and clinical characteristics of these two groups. The primary tumor sites were distributed as follows: 3273 cases (47.0%) in the oral cavity, 1790 cases (25.7%) in the oropharynx, 1489 cases (21.4%) in the hypopharynx, and 407 cases (5.8%) in the larynx. The T stage distribution showed T4 as the most common with 3617 cases (52.0%), followed by T2 with 1741 cases (25.0%). For N stages, N2 was most prevalent with 3352 cases (48.2%), followed by N0 with 1969 cases (28.3%). According to the AJCC 7th edition, stage 4 was the most prevalent with 5559 cases (79.7%), followed by stage 3 with 752 cases (10.8%).

Patients in the pneumonia group were older, with a median age of 56.2 years compared to 53.3 years in the non-pneumonia group (*p* < 0.001). They also had a higher incidence of diabetes mellitus (9.7% vs. 6.4%, *p* < 0.001), and underwent more invasive procedures, such as tracheostomy (35.2% vs. 30.9%, *p* = 0.004), gastrostomy (6.6% vs. 3.5%, *p* < 0.001), and nasogastric tube placement (76.5% vs. 67.4%, *p* < 0.001). Additionally, these patients exhibited poorer nutrition status, with a pre-treatment weight of 60 kg compared to 64 kg (*p* < 0.001) and a post-treatment weight of 56 kg compared to 60 kg (*p* < 0.001). They experienced longer durations of chemotherapy (77 days vs. 56 days, *p* < 0.001) and radiotherapy (50 days vs. 49 days, *p* < 0.001), as well as more frequent delays of at least two weeks in radiotherapy (14.7% vs. 7.9%, *p* < 0.001).

### 3.2. Correlation between Survival and Pneumonia

Our study delved into the relationship between OS and the incidence of pneumonia in HNSCC patients treated with CCRT. As the Kaplan–Meier curve revealed in Figure 2, we identified statistically significant disparities in median OS between the groups afflicted with pneumonia and those without (1.2 years vs. 4.9 years, adjusted HR 1.900, 95% CI 1.770 to 2.040, Log-rank *p*-value < 0.001). The 2-year (68.2% vs. 36.1%), 5-year (49.5% vs. 23.2%), and 10-year (32.4% vs. 14.2%) OS rates exhibited notable discrepancies between the non-pneumonia and pneumonia groups, with statistically significant differences observed.

Table 2 shows the OS risk assessment using univariate analysis with a further multivariate Cox proportional hazards model and revealed independent prognostic factors in HNSCC patients undergoing CCRT: age (adjusted HR 1.008), male sex (adjusted HR 1.282), with pneumonia (adjusted HR 1.900), alcohol consumption(adjusted HR 1.151), advanced N-classification (N2, adjusted HR 1.577; N3, adjusted HR 2.323), history of DM (adjusted HR 1.436), having a gastrostomy (adjusted HR 1.481), using an NG tube (adjusted HR 1.215), longer duration of chemotherapy (adjusted HR 1.003), and RT delay of at least two weeks (adjusted HR 1.949). The primary tumor site over the larynx (adjusted HR 0.797) showed a better prognosis in this study.

### 3.3. Risk Factors for Pneumonia in HNSCC Patients with CCRT

Due to the higher mortality rate observed within the pneumonia group, logistic regression analysis was employed to delve deeper into the risk factors associated with pneumonia among HNSCC patients undergoing CCRT (Table 3). Following univariate and subsequent multivariate analysis, the independent risk factors identified within the pneumonia group included older age (adjusted OR 1.037), male sex (adjusted OR 1.348), alcohol consumption (adjusted OR 1.220), advanced N-classification (N2, adjusted OR 1.307; N3, adjusted OR 1.511), history of DM (adjusted OR 1.457), having a gastrostomy (adjusted OR 1.342), using an NG tube (adjusted OR 1.764), longer duration of chemotherapy (adjusted OR 1.006), radiotherapy delay of at least 2 weeks (adjusted OR 1.795), and primary tumor site over hypopharynx (Larynx, adjusted OR 0.664; Oral cavity, adjusted OR 0.686; Oropharynx, adjusted OR 0.734).

Then we used Greedy propensity score matching 1:2 (based on age, AJCC stage, and primary site) to reduce the bias (Table 4) further. Following univariate and subsequent multivariate analysis, the independent risk factors identified within the pneumonia group included alcohol consumption (adjusted OR 1.237), history of DM (adjusted OR 1.511), having a gastrostomy (adjusted OR 1.354), using an NG tube (adjusted OR 1.722), longer duration of chemotherapy (adjusted OR 1.005), and a radiotherapy delay of at least two weeks (adjusted OR 1.752).

## 4. Discussion

CCRT aims to achieve curative outcomes in patients diagnosed with locally advanced HNSCC. However, the occurrence of pneumonia during CCRT often disrupts radiotherapy unexpectedly, leading to diminished cure rates, shortened duration of remission, and compromised survival outcomes [12,14]. Our current study showed the HNSCC patient had a high risk of pneumonia during CCRT. Previous studies showed varied rates of aspiration pneumonia across different HNSCC populations. Mortensen et al. documented that 5.3% of patients diagnosed with HNSCC experienced aspiration pneumonia within the initial year following radiotherapy, whether administered alone or concurrently with chemotherapy. This finding aligns closely with the results of Chun et al. In a retrospective study conducted in Taiwan, encompassing 15,894 patients diagnosed with HNSCC and relying on health insurance claims data, a pneumonia incidence of 5% was observed during the course of radiotherapy. However, while HNSCC received CCRT, Hunter et al. reported 15%, and Eisbruch et al. reported 23% aspiration pneumonia rates. The limitations of these studies lie in the small number of patients enrolled. To our knowledge, this multicenter retrospective study represents one of the largest cohorts (N = 6959) to date, focusing specifically on the impact of pneumonia on survival outcomes among patients with HNSCC undergoing CCRT in regions endemic for HNSCC. The incidence of pneumonia in our study was 23.2%.

The increased risk of pneumonia during CCRT substantially impacts HNSCC patients’ survival. The median OS in our study showed statistically significant differences between pneumonia and non-pneumonia groups (1.2 years vs. 4.9 years, HR: 1.900). Xu et al. reported similar findings, demonstrating that the 5-year survival rate from diagnosis was 15.2% for those with aspiration pneumonia, compared to 32.3% for those without aspiration pneumonia. However, a post hoc analysis from a phase 3 study enrolling 536 patients with HNSCC receiving CCRT in India showed no statistically significant differences between aspiration pneumonia (AsP) and non-AsP patient (35.9 months vs. 47.1 months, *p* = 0.13, HR for death 1.233, 95% CI 0.939–1.618) [13]. The reason for the discrepancy may be different enrollment populations. Our study included more stage IV patients (86.1% in pneumonia group, 78.0% in the non-pneumonia group) compared to Vijay Patil et al. (30.4% in pneumonia group, 69.6% in the non-pneumonia group). Moreover, in their study, there were no differences of age between the pneumonia group and the non-pneumonia group. However, the age in the pneumonia group was older than the non-pneumonia group in our study.

Previous studies have focused on the early mortality rate following the initiation of CCRT. The risk of death within 30 days after completing CCRT is 5.4%, and within 60 days, it rises to 7.2% [15,16]. Our study indicated that HNSCC patients who developed pneumonia had a higher mortality rate not only during the CCRT period but also extending through a 10-year follow-up period. Jeffrey et al. also showed a 49% overall survival rate in CCRT for locally advanced HNSCC patients after 10 years of follow-up [17]. Their 10-year survival rate was higher than ours because our study had more stage IV patients.

An interesting discovery in our present study for HNSCC patients receiving CCRT is the significant increase in pneumonia risk associated with gastrostomy and nasogastric tube feeding. Complications during CCRT, such as mucositis, dehydration, and xerostomia, often disrupt swallowing function and increase the risks of malnutrition and aspiration pneumonia. To overcome malnourishment in HNSCC patients, establishing a proper feeding circuit is necessary, including gastrostomy and nasogastric tubes. A prophylactic gastrostomy and nasogastric tube before CCRT may benefit patients with difficulty eating. Recently, the prophylactic utilization of gastrostomy or nasogastric tubes for feeding gained a lot of attention in HNSCC patients. Kano et al. reported that the gastrostomy tube group exhibited superiority over the nasogastric tube group regarding the incidence of aspiration pneumonia (13% vs. 43%), the completion rate of CCRT (79% vs. 62%), and duration of hospitalization post-CCRT (24 days vs. 36 days) [18]. Nevertheless, a systemic review from Maurizio et al. found that neither gastrostomy nor nasogastric tubes offer advantages in nutrition status, radiation interruption, or survival [19]. The possible reason for the difference between Kano et al. and our study may be the selection bias. The different conclusions of the two studies might come from different proportions of intervention preference. Very few people have gastrostomy (4.2%), but many have nasogastric tubes (69.5%) in our study, compared to fewer people who have nasogastric tubes (14%) but many who have gastrostomy (71%) in Kano et al.’s study. Prophylactic gastrostomy or nasogastric tube often was suggested in poor prognostic groups, such as old age, poor performance status, and advanced tumor stage [18,20].

Our study also showed older age and advanced tumor stage (T classification, N classification, AJCC tumor stage) predicted poor OS in HNSCC receiving CCRT. Lin et al. revealed that elderly patients with locally advanced HNSCC undergoing CCRT exhibited a higher 90-day mortality rate following the completion of radiotherapy compared to younger patients [21]. Our study found that older age and male gender were positively correlated with the occurrence of pneumonia during CCRT. Xu et al. and Chu et al. also identified these pneumonia risk factors in HNSCC patients. The aspiration pneumonia rate is higher in elderly cancer and non-cancer patients. Aspiration pneumonia may arise from age-related decline in protective swallowing reflexes [22]. Evidence also suggests that HNSCC with higher cytotoxic CD8+ T cell infiltration has a better prognosis, but the infiltration of cytotoxic CD8+ T cells decreases with aging. Bronchoscopic lavage examinations have also shown that CD8+ T cell infiltration in the lungs decreases with aging. This evidence indicates that immunosenescence may contribute to the increased incidence of HNSCC and the higher risk of lung infections with advancing aging.

Gender as a risk factor remains controversial in previous research [10,12,13]. In our study, the male sex was a significantly poorer risk factor for survival but was not significant for pneumonia. Alcohol consumption was a significant risk factor in our study, consistent with Géraldine Descamps et al., who revealed the risk of mortality is markedly elevated among HNSCC patients who smoke tobacco and consume alcohol during treatment [23]. The markedly higher incidence of HNSCC in males compared to females in Taiwan is primarily attributed to the substantially greater prevalence of alcohol abuse, tobacco smoking, and betel nut chewing among men. And these risk factors also contribute to the poorer prognosis observed in male patients.

Diabetes mellitus (DM), a part of the Charlson comorbidity index (CCI), is increasing in incidence in Taiwan [24]. The association between DM and HNSCC was reported in several previous studies [25,26]. Our study revealed a significant increase in the risk of pneumonia and mortality in HNSCC patients receiving CCRT when accompanied by a diagnosis of DM. Kuo et al. also showed patients with HNSCC and DM undergoing CCRT experienced higher rates of infection and higher treatment-related mortality [27]. However, the previous study seldom explored the relationship between DM and pneumonia in HNSCC patients undergoing CCRT. Our study proves that HNSCC patients with DM should take care of pneumonia while they receive CCRT.

Head and neck cancer patients receive elective tracheostomy while airway obstruction is anticipated. One study demonstrated evidence of aspiration pneumonia in 69% of a cohort comprising surgical patients with tracheostomies [28]. However, few studies investigated the correlation between tracheostomy and pneumonia in HNSCC patients with CCRT. Our study showed that tracheostomy did not increase the risk of pneumonia and impact survival. This result is consistent with Gina et al., who revealed that tracheotomy dependence did not demonstrate any impact on local control, progression-free survival, or overall survival in 109 HNSCC patients treated with chemoradiation [29].

Finally, a longer duration of chemotherapy and a delay of at least two weeks in radiotherapy independently emerged as risk factors for pneumonia in HNSCC patients with CCRT and predicted inferior outcomes. Sher et al. found that a prolonged radiotherapy duration of more than 51 days was associated with low overall survival [30]. Some studies showed pneumonia during CCRT may prolong the overall treatment time, leading to a decrease in locoregional control and, thus, an increase in mortality [10,12]. As above, developing pneumonia around the time of diagnosis is associated with a longer duration of chemotherapy and a higher rate of radiotherapy delay. This may be due to treatment interruptions caused by pneumonia, which could reduce treatment efficacy and lead to a lower overall survival rate over long-term follow-up. Furthermore, patients with more comorbidities or complications from HNSCC, such as diabetes mellitus or anatomical abnormalities due to the cancer or its treatments (e.g., tracheostomy or nasogastric tube placement), have weakened immunity, making them more susceptible to pneumonia. Consequently, the lower long-term survival rate among pneumonia patients might be also because those who develop pneumonia tend to have a poorer prognosis due to the underlying severity of their disease.

### Strengths and Limitations

Our methodological approach demonstrates several strengths, particularly the incorporation of a substantial sample size, the utilization of real-world clinical data, a multicenter study approach, and an adequate follow-up period that facilitates identification of differences in survival between groups.

This research is subject to several limitations. First, study data were extracted from the CGRD, established by the hospitals of a single system in Taiwan. The treatment methods, including chemotherapy, radiotherapy, and supportive care, may be different from other hospitals. Therefore, our findings face challenges in terms of external verification by other researchers. Second, the population of our study is almost all Asian. Asians seem to have better OS than non-Asians in HNSCC [31], and a more significant proportion of non-oropharyngeal cancer is diagnosed by age 40 compared to the non-Hispanic White population [32]. Therefore, this study limited the credibility of extending the findings to the non-Hispanic White population. Third, further analysis of radiation dose and chemotherapy regimen in CCRT was not performed in our study. Further research with sub-analyses on these topics is required. Lastly, this retrospective study offered a lower level of evidence compared to prospective studies due to their inherent limitations in study design, such as reliance on existing data and potential for bias. Therefore, we are looking forward to a well-designed prospective study.

## 5. Conclusions

This retrospective, multicenter study, comprising a considerable sample size of 6959 participants, investigates the survival outcomes and identifies risk factors associated with pneumonia in patients with HNSCC following CCRT. Pneumonia compromised the survival in HNSCC patients with CCRT. HNSCC patients undergoing CCRT are at an elevated risk of pneumonia and mortality due to factors such as alcohol consumption, history of diabetes mellitus, gastrostomy or nasogastric tube placement, prolonged chemotherapy duration, or radiotherapy delay of at least 2 weeks.

## Figures and Tables

**Figure 1 biomedicines-12-01480-f001:**
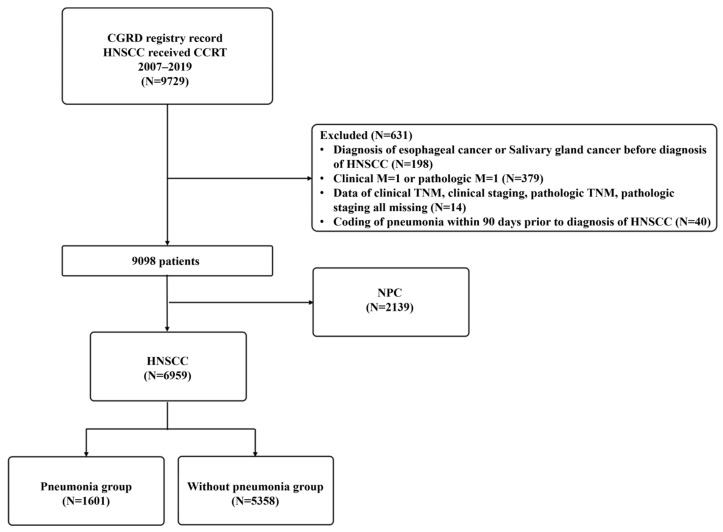
Flow chart depicting the criteria for inclusion and exclusion of HNSCC patients undergoing CCRT enrolled in this study. In this study, a total of 6959 patients diagnosed with HNSCC were included and categorized into two groups: group 1 comprised patients with pneumonia (N = 1601), while group 2 consisted of patients without pneumonia (N = 5358).

**Figure 2 biomedicines-12-01480-f002:**
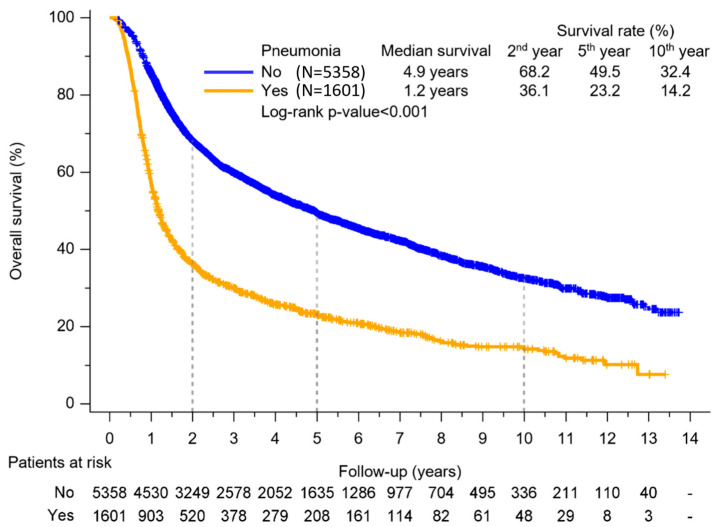
Kaplan–Meier method for OS of HNSCC patients treated with CCRT. Log-rank test assessed the overall survival difference between groups with pneumonia and without pneumonia. Relative to the non-pneumonic group, individuals with HNSCC treated with CCRT displayed notably diminished 2-year, 5-year, and 10-year survival rates in the pneumonia group.

**Table 1 biomedicines-12-01480-t001:** Demographic and clinical characteristics of this study. Data are n (%), unless otherwise indicated. Age is presented as the mean ± SD. Other continuous variables are presented as median (IQR). The categories with statistically significant differences between the pneumonia and non-pneumonia groups include age, history of diabetes, tracheostomy, gastrostomy, nasogastric tube, pre-treatment weight, post-treatment weight, duration of chemotherapy, duration of radiotherapy, and radiotherapy delay of at least two weeks. Abbreviations: SD, standard deviation; IQR, interquartile range; AJCC, American Joint Committee on Cancer; HNSCC, squamous cell carcinoma of head and neck; CCRT, concurrent chemoradiotherapy.

Variables	Total N = 6959	Pneumonia N = 1601 (23.01)	Without Pneumonia N = 5358 (76.99)	*p*-Value
Age (years)	53.9 ± 10.1	56.2 ± 10.5	53.3 ± 9.9	<0.001
Gender				0.009
Male	6550 (94.1)	1529 (95.5)	5021 (93.7)	
Female	409 (5.9)	72 (4.5)	337 (6.3)	
Cigarette smoking				0.303
Never	672 (9.7)	137 (8.6)	535 (10.0)	
Former or current	4475 (64.3)	1050 (65.6)	3425 (63.9)	
Missing	1812 (26.0)	414 (25.9)	1398 (26.1)	
Alcohol consumption				0.069
Never	1207 (17.3)	248 (15.5)	959 (17.9)	
Former or current	3915 (56.3)	937 (58.5)	2978 (55.6)	
Missing	1837 (26.4)	416 (26.0)	1421 (26.5)	
Betel nuts chewing				0.957
Never	1418 (20.4)	324 (20.2)	1094 (20.4)	
Former or current	3728 (53.6)	863 (53.9)	2865 (53.5)	
Missing	1813 (26.1)	414 (25.9)	1399 (26.1)	
Primary tumor site				<0.001
Hypopharynx	1489 (21.4)	439 (27.4)	1050 (19.6)	
Larynx	407 (5.8)	82 (5.1)	325 (6.1)	
Oral Cavity	3273 (47.0)	697 (43.5)	2576 (48.1)	
Oropharynx	1790 (25.7)	383 (23.9)	1407 (26.3)	
T classification				<0.001
0	72 (1.0)	9 (0.6)	63 (1.2)	
1	499 (7.2)	80 (5.0)	419 (7.8)	
2	1741 (25.0)	310 (19.4)	1431 (26.7)	
3	1030 (14.8)	241 (15.1)	789 (14.7)	
4	3617 (52.0)	961 (60.0)	2656 (49.6)	
N classification				<0.001
0	1969 (28.3)	370 (23.1)	1599 (29.8)	
1	997 (14.3)	191 (11.9)	806 (15.0)	
2	3352 (48.2)	842 (52.6)	2510 (46.8)	
3	641 (9.2)	198 (12.4)	443 (8.3)	
Stage (AJCC, 7th ed.)				<0.001
0 + 1	151 (2.2)	23 (1.4)	128 (2.4)	
2	497 (7.1)	66 (4.1)	431 (8.0)	
3	752 (10.8)	134 (8.4)	618 (11.5)	
4	5559 (79.9)	1378 (86.1)	4181 (78.0)	
Diabetes mellitus				<0.001
No	6461 (92.8)	1446 (90.3)	5015 (93.6)	
Yes	498 (7.2)	155 (9.7)	343 (6.4)	
Tracheostomy				0.004
No	4739 (68.1)	1037 (64.8)	3702 (69.1)	
Yes	2220 (31.9)	564 (35.2)	1656 (30.9)	
Gastrostomy				<0.001
No	6669 (95.8)	1496 (93.4)	5173 (96.5)	
Yes	290 (4.2)	105 (6.6)	185 (3.5)	
Nasogastric tube				<0.001
No	2125 (30.5)	376 (23.5)	1749 (32.6)	
Yes	4834 (69.5)	1225 (76.5)	3609 (67.4)	
Esophageal cancer incidence				0.420
No	6526 (93.8)	1507 (94.1)	5019 (93.7)	
Yes	433 (6.2)	94 (5.9)	339 (6.3)	
Duration of chemotherapy (days)	58 (39–114)	77 (42–175)	56 (36–100)	<0.001
Duration of radiotherapy (days)	49 (46–54)	50 (46–57)	49 (45–53)	<0.001
Albumin (g/dL)	4.26 (3.9–4.51)	4.1 (3.6–4.4)	4.3 (3.97–4.55)	<0.001
Body weight-pre (kg)	63 (55.5–71)	60 (53–68.6)	64 (56.5–72)	<0.001
Body weight-post (kg)	59 (52–66)	56 (50–63)	60 (53–67)	<0.001
body weight loss (kg)	4 (1–7)	4 (0.1–8)	4 (1–7)	0.901
Radiotherapy delay of at least 2 weeks				<0.001
No	6301 (90.5)	1366 (85.3)	4935 (92.1)	
Yes	658 (9.5)	235 (14.7)	423 (7.9)	
Interval between diagnosis of HNSCC and initiation of CCRT (days)	42 (775)	39 (744)	42 (527)	0.115

**Table 2 biomedicines-12-01480-t002:** Univariate and multivariate analysis for Cox proportional hazard model of prognostic factors in HNSCC patients with CCRT. The categories that remain statistically significant after univariate and multivariate analysis include age, sex, pneumonia, alcohol consumption, N-classification 2 and 3, stage 4, diabetes mellitus, gastrostomy, nasogastric tube, duration of chemotherapy, radiotherapy delay of at least two weeks, and primary tumor site in the larynx. Abbreviations: HNSCC, Squamous cell carcinoma of head and neck; CCRT, Concurrent chemoradiotherapy; HR, Hazard ratio; C.I., Confidence interval; AJCC, American Joint Committee on Cancer.

Variables	Crude HR (95% C.I.)	*p*-Value	Adjusted HR (95% C.I.)	*p*-Value
Age	1.008 (1.005, 1.011)	<0.001	1.008 (1.005, 1.012)	<0.001
Gender				
Male	1.460 (1.260, 1.693)	<0.001	1.282 (1.097, 1.499)	0.002
Female	Reference		Reference	
Pneumonia				
No	Reference		Reference	
Yes	2.353 (2.200, 2.517)	<0.001	1.900 (1.770, 2.040)	<0.001
Cigarette smoking				
Never	Reference		Reference	
Former or current	1.176 (1.045, 1.324)	0.007	1.071 (0.933, 1.229)	0.329
Alcohol consumption				
Never	Reference		Reference	
Former or current	1.194 (1.087, 1.311)	<0.001	1.151 (1.034, 1.281)	0.010
Betel nuts chewing				
Never	Reference		Reference	
Former or current	1.026 (0.942, 1.118)	0.552	0.856 (0.776, 0.944)	0.002
T classification				
0	Reference		Reference	
1	1.009 (0.701, 1.452)	0.962	0.660 (0.453, 0.963)	0.031
2	1.087 (0.768, 1.539)	0.638	0.749 (0.524, 1.070)	0.112
3	1.413 (0.994, 2.007)	0.054	0.975 (0.681, 1.395)	0.889
4	1.893 (1.343, 2.669)	<0.001	1.281 (0.905, 1.813)	0.162
N classification				
0	Reference		Reference	
1	1.050 (0.944, 1.168)	0.370	1.194 (1.058, 1.347)	0.004
2	1.498 (1.389, 1.616)	<0.001	1.577 (1.436, 1.731)	<0.001
3	2.421 (2.157, 2.716)	<0.001	2.323 (2.039, 2.646)	<0.001
Stage (AJCC, 7th ed.)				
0 + 1	Reference		Reference	
2	0.964 (0.726, 1.280)	0.799	0.901 (0.656, 1.237)	0.518
3	0.961 (0.732, 1.260)	0.771	0.661 (0.487, 0.898)	0.008
4	1.693 (1.317, 2.177)	<0.001	0.630 (0.470, 0.844)	0.002
Diabetes mellitus				
No	Reference		Reference	
Yes	1.370 (1.224, 1.533)	<0.001	1.436 (1.278, 1.613)	<0.001
Tracheostomy				
No	Reference		Reference	
Yes	1.056 (0.988, 1.127)	0.107	0.948 (0.883, 1.019)	0.145
Gastrostomy				
No	Reference		Reference	
Yes	2.004 (1.756, 2.288)	<0.001	1.481 (1.291, 1.699)	<0.001
Nasogastric tube				
No	Reference		Reference	
Yes	1.258 (1.174, 1.348)	<0.001	1.215 (1.128, 1.308)	<0.001
Duration of chemotherapy	1.004 (1.004, 1.005)	<0.001	1.003 (1.003, 1.003)	<0.001
Duration of radiotherapy	1.001 (0.998, 1.003)	0.527	0.983 (0.980, 0.986)	<0.001
Radiotherapy delay of at least 2 weeks				
No	Reference		Reference	
Yes	1.658 (1.509, 1.821)	<0.001	1.949 (1.716, 2.215)	<0.001
Primary tumor site				
Hypopharynx	Reference		Reference	
Larynx	0.603 (0.512, 0.709)	<0.001	0.797 (0.674, 0.942)	0.008
Oral Cavity	0.821 (0.760, 0.887)	<0.001	0.998 (0.916, 1.088)	0.969
Oropharynx	0.823 (0.754, 0.898)	<0.001	0.964 (0.882, 1.055)	0.429
Esophageal cancer incidence				
No	Reference		Reference	
Yes	1.175 (1.048, 1.318)	0.006	1.111 (0.988, 1.249)	0.080

**Table 3 biomedicines-12-01480-t003:** Univariate and multivariate analysis for logistic regression of risk factors for pneumonia in HNSCC patients with CCRT (No matching). The categories that remain statistically significant after univariate and multivariate analysis include age, gender, alcohol consumption, N-classification 2 and 3, diabetes mellitus, gastrostomy, nasogastric tube, duration of chemotherapy, radiotherapy delay of at least two weeks, and primary tumor site in the hypopharynx. Abbreviations: HNSCC, Squamous cell carcinoma of head and neck; CCRT, Concurrent chemoradiotherapy; OR, Odds ratio; C.I., Confidence interval; AJCC, American Joint Committee on Cancer.

Variables	Crude OR (95% C.I.)	*p*-Value	Adjusted OR (95% C.I.)	*p*-Value
Age (years)	1.028 (1.023, 1.034)	<0.001	1.037 (1.031, 1.044)	<0.001
Gender				
Male	1.425 (1.098, 1.849)	0.008	1.348 (1.016, 1.788)	0.038
Female	Reference		Reference	
Cigarette smoking				
Never	Reference		Reference	
Former or current	1.197 (0.980, 1.462)	0.078	1.085 (0.854, 1.378)	0.504
Alcohol consumption				
Never	Reference		Reference	
Former or current	1.217 (1.039, 1.425)	0.015	1.220 (1.015, 1.466)	0.034
Betel nuts chewing				
Never	Reference		Reference	
Former or current	1.017 (0.879, 1.176)	0.819	0.909 (0.766, 1.078)	0.273
T classification				
0	Reference		Reference	
1	1.336 (0.639, 2.796)	0.441	1.077 (0.497, 2.337)	0.850
2	1.516 (0.746, 3.082)	0.250	1.418 (0.676, 2.972)	0.355
3	2.138 (1.048, 4.363)	0.037	1.694 (0.807, 3.557)	0.164
4	2.533 (1.255, 5.112)	0.010	1.958 (0.948, 4.043)	0.069
N classification				
0	Reference		Reference	
1	1.024 (0.843, 1.243)	0.810	1.021 (0.818, 1.275)	0.852
2	1.450 (1.264, 1.663)	<0.001	1.307 (1.097, 1.558)	0.003
3	1.932 (1.578, 2.365)	<0.001	1.511 (1.193, 1.914)	0.001
Stage (AJCC, 7th ed.)				
0 + 1	Reference		Reference	
2	0.852 (0.510, 1.425)	0.542	0.675 (0.377, 1.210)	0.187
3	1.207 (0.746, 1.953)	0.444	0.802 (0.463, 1.391)	0.432
4	1.834 (1.172, 2.871)	0.008	0.754 (0.444, 1.282)	0.297
Diabetes mellitus				
No	Reference		Reference	
Yes	1.568 (1.286, 1.912)	<0.001	1.457 (1.178, 1.801)	0.001
Tracheostomy				
No	Reference		Reference	
Yes	1.216 (1.081, 1.368)	0.001	1.129 (0.988, 1.292)	0.075
Gastrostomy				
No	Reference		Reference	
Yes	1.963 (1.534, 2.511)	<0.001	1.342 (1.027, 1.753)	0.031
Nasogastric tube				
No	Reference		Reference	
Yes	1.578 (1.388, 1.796)	<0.001	1.764 (1.531, 2.031)	<0.001
Duration of chemotherapy	1.005 (1.005, 1.006)	<0.001	1.006 (1.005, 1.006)	<0.001
Duration of radiotherapy	1.008 (1.004, 1.012)	<0.001	0.996 (0.991, 1.000)	0.065
Radiotherapy delay of at least 2 weeks				
No	Reference		Reference	
Yes	2.007 (1.693, 2.380)	<0.001	1.795 (1.419, 2.271)	<0.001
Primary tumor site				
Hypopharynx	Reference		Reference	
Larynx	0.603 (0.462, 0.788)	<0.001	0.664 (0.498, 0.885)	0.005
Oral Cavity	0.647 (0.563, 0.744)	<0.001	0.686 (0.585, 0.805)	<0.001
Oropharynx	0.651 (0.555, 0.763)	<0.001	0.734 (0.619, 0.870)	<0.001
Esophageal cancer incidence				
No	Reference		Reference	
Yes	0.923 (0.730, 1.169)	0.508	0.817 (0.636, 1.051)	0.116

**Table 4 biomedicines-12-01480-t004:** After Greedy propensity score matching 1:2, univariate and multivariate analysis with logistic regression of risk factors for pneumonia in HNSCC patients with CCRT. The categories that remain statistically significant after univariate and multivariate analysis include alcohol consumption, diabetes mellitus, gastrostomy, nasogastric tube, duration of chemotherapy, and radiotherapy delay of at least two weeks. Abbreviations: HNSCC, Squamous cell carcinoma of head and neck; CCRT, Concurrent chemoradiotherapy; OR, Odds ratio; C.I., Confidence interval.

Variables	Crude OR (95% C.I.)	*p*-Value	Adjusted OR (95% C.I.)	*p*-Value
Gender				
Male	1.306 (0.987, 1.728)	0.062	1.206 (0.896, 1.622)	0.217
Female	Reference		Reference	
Cigarette smoking				
Never	Reference		Reference	
Former or current	1.262 (1.018, 1.565)	0.034	1.084 (0.841, 1.397)	0.536
Alcohol consumption				
Never	Reference		Reference	
Former or current	1.216 (1.024, 1.443)	0.025	1.237 (1.019, 1.501)	0.032
Betel nuts chewing				
Never	Reference		Reference	
Former or current	1.010 (0.862, 1.183)	0.901	0.840 (0.701, 1.005)	0.057
T classification				
0	Reference		Reference	
1	1.498 (0.693, 3.241)	0.305	1.412 (0.641, 3.108)	0.392
2	1.884 (0.900, 3.945)	0.093	1.762 (0.829, 3.747)	0.141
3	2.221 (1.056, 4.673)	0.035	1.948 (0.912, 4.157)	0.085
4	2.372 (1.142, 4.926)	0.021	1.940 (0.922, 4.082)	0.081
N classification				
0	Reference		Reference	
1	1.020 (0.824, 1.263)	0.853	0.984 (0.790, 1.226)	0.889
2	1.198 (1.032, 1.392)	0.018	1.060 (0.906, 1.241)	0.468
3	1.439 (1.158, 1.788)	0.001	1.174(0.934, 1.474)	0.169
Diabetes mellitus				
No	Reference		Reference	
Yes	1.352 (1.089, 1.679)	0.006	1.511 (1.206, 1.892)	<0.001
Tracheostomy				
No	Reference		Reference	
Yes	1.195 (1.052, 1.358)	0.006	1.078 (0.939, 1.237)	0.285
Gastrostomy				
No	Reference		Reference	
Yes	1.773 (1.354, 2.322)	<0.001	1.354 (1.017, 1.803)	0.038
Nasogastric tube				
No	Reference		Reference	
Yes	1.639 (1.426, 1.883)	<0.001	1.722 (1.489, 1.992)	<0.001
Duration of chemotherapy	1.005 (1.004, 1.006)	<0.001	1.005 (1.004, 1.006)	<0.001
Duration of radiotherapy	1.006 (1.002, 1.010)	0.001	0.995 (0.990, 1.000)	0.057
Radiotherapy delay of at least 2 weeks				
No	Reference		Reference	
Yes	1.741 (1.445, 2.098)	<0.001	1.752 (1.363, 2.253)	<0.001
Esophageal cancer incidence				
No	Reference		Reference	
Yes	0.888 (0.690, 1.143)	0.356	0.835 (0.643, 1.085)	0.177

## Data Availability

Access to these data is restricted. The data were sourced from the Chang Gung Research Database and can be made available upon approval from the IRB of the Kaohsiung and Chiayi branches of Chang Gung Memorial Hospital.

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
