# Peer review of "Prognosis of Pneumonia in Head and Neck Squamous Cell Carcinoma Patients Who Received Concurrent Chemoradiotherapy"

_biomedicines, 2024, doi:10.3390/biomedicines12071480_

Round 1
Reviewer 1 Report
Comments and Suggestions for Authors
This study investigated the survival status and associated risk factors for pneumonia in patients with head and neck squamous cell carcinoma who underwent CCRT, and identifies some risk factors. This paper provides useful information for doctors who treat patients with head and neck squamous cell carcinoma, and is worthy of publication in an academic journal. However, some minor revisions are necessary, so please review the following.
Please spell out the following abbreviations when they first appear in the text, and indicate that the abbreviation is used in the text.
“CCRT, CGRD, DM, OS, CCI”
Line 157-158
The number of people in each group listed in the text is different from that in Figure 1. Please correct the number.
Line 175-182
There are several differences between the number and percentage in Table 1 and those listed in the text. Please write the correct number.
Line 176
Regarding “more comorbidities”
This expression is inappropriate because only DM was investigated.
Line 178
Regarding “poor nutrition”
It is inappropriate to evaluate nutritional status by weight. BMI should be used rather than weight for comparison.
Line 316
“betel nut consumption”
Please provide evidence that this affects the risk of death.
Author Response
Comments 1: Please spell out the following abbreviations when they first appear in the text, and indicate that the abbreviation is used in the text.
“CCRT, CGRD, DM, OS, CCI”
Response 1:
Thank you for pointing this out. We agree with this comment. Therefore, we have added concurrent chemoradiotherapy (CCRT) at line 55, added Chang Gung Research Database (CGRD) at line 90, added diabetes mellitus (DM) at line 129, added overall survival (OS) at line 139, and added Charlson comorbidity index (CCI) at line 364.
Comments 2: Line 157-158
The number of people in each group listed in the text is different from that in Figure 1. Please correct the number.
Response 2:
Thank you so much for pointing this out. We have, accordingly, corrected the number in the text to match the number in Figure 1. We corrected 1625 (23.2%) to 1601 (23.01%) at line 32 and line 162, and corrected 5367 (76.8%) to 5358 (76.99%) at line 162.
Comments 3: Line 175-182
There are several differences between the number and percentage in Table 1 and those listed in the text. Please write the correct number.
Response 3:
Thank you for your careful reminder. We have reviewed all the numbers in the tables and text to ensure their accuracy.
Comments 4: Line 176
Regarding “more comorbidities”
This expression is inappropriate because only DM was investigated.
Response 4:
We changed the text “had more comorbidities (DM…)” to “had a higher incidence of diabetes mellitus” at line 201. Thanks for your careful reminder.
Comments 5: Line 178
Regarding “poor nutrition”
It is inappropriate to evaluate nutritional status by weight. BMI should be used rather than weight for comparison.
Response 5:
BMI is a good assessment tool for evaluating nutrition status in cancer patients. However, it isn't easy to compare the differences during a short time of treatment. In this study, we are concerned about significant weight loss over a short period and any noticeable weight changes during follow-up after treatment. These changes might indicate treatment side effects or disease of head and neck cancer, both of which are related to malnutrition.
Comments 6: Line 316
“betel nut consumption”
Please provide evidence that this affects the risk of death.
Response 6:
According to the following reference, betel nut chewing alone significantly increases the risk of overall mortality (adjusted IRR = 1.44, 95% CI = 1.27-1.63) and cancer-specific mortality (adjusted IRR = 1.51, 95% CI = 1.30-1.44), regardless of gender or location of head and neck cancer. Furthermore, a history of betel nut chewing was independently associated with an inferior progression-free survival rate (2-year PFS: 37% vs. 67%, p = 0.004) and a poor overall survival rate (2-year OS: 47% vs. 71%, p = 0.017).
Reference:
Su, YY., Chien, CY., Luo, SD. et al. Betel nut chewing history is an independent prognosticator for smoking patients with locally advanced stage IV head and neck squamous cell carcinoma receiving induction chemotherapy with docetaxel, cisplatin, and fluorouracil. World J Surg Onc 14, 86 (2016).
Su MJ, Ho CH, Yeh CC. Association of alcohol consumption, betel nut chewing, and cigarette smoking with mortality in patients with head and neck cancer among the Taiwanese population: A nationwide population-based cohort study. Cancer Epidemiol. 2024;89:102526.
Reviewer 2 Report
Comments and Suggestions for Authors
In this retrospective, multicenter study data of more than 6900 patients was analyzed regarding the prognosis of pneumonia in HNSCC receiving concurrent chemoradiotherapy.
I enjoyed reading the study. However I have a few major comments to make:
Abstract:
1. Ll 31-32: 1625 of 6959 patients is 23.4 %
2. Line 34: What do you mean with „more diabetes mellitus“? Patients who suffered from DM or patients who suffered more severely from DM?
Introduction:
3. Line 45: Please delete cavity
4. Line 57: Do you mean „local recurrence“ with „local failure“?
5. Line 61: Please write „difficulties“
Material and Methods:
6. All numbers that are mentioned should be mentioned in the results section. The explanations given could remain in the Material and Methods section.
7. Line 133: Please write „independent“^
Results:
8. Line 157: 23,4% see comment above.
9. Line 158: 5367 of 6959 patients are 77.1%, not 76.8%. In figure 1 it is written that 5358 patients did not have a penumonia. Which number is correct?
10. Ll: 161-162: Please add the numbers of patients to the given percentages.
11. Table 1:
I. Your cohort consists of only 5.9% females. It is known that males suffer more often from HNSCC. However, do you have any explanations for this major gender imbalance?
II. Do you also collected pack years?
III. Please write „Nasogastric“
IV. Please explain what is written in the table: data is shown in number of patients and percentages (you might delete „%“ in each cell), median with minimum and maximum or mean with SD?
V. Radiotherapy delay of more than 2 weeks?
VI. A legend with the main results of this table and explanation of the given numbers is missing. Please add.
12. Ll 181 and 337l: Delay of at least 2 weeks?
13. Line 182: A fullstop is missing.
14. Ll 175-182: This sentence is pretty long and hard to read.
15. Line 185: Please write curve
16. Lines 200 and 217: What do you mean with „more alcohol drinking“/ „more alcohol consumption“? In table 1 and 2 you classified „never“ „former or current“
17. Table 2-4: Llegends are missing
18. Ll 214-236: This passage is hard to read and numbers are not the exactly the same like in table 3. Please correct the numbers in the table and delete them in the text as they are written in the table.
19. Ll 232-238: Please delete the numbers in the text.
Discussion
20. Line 258: What do you mean with „limited“?
21. Ll 272-274: a verb is missing
22. Line 275: Please write „was“.
23. Ll 276-282: Do the authors have any idea how having suffered from pneumonia around the time of diagnosis could influence the 10-year survival rates?
24. Ll 304-319: In general the aging immune system loses the ability to protect against infections and cancer. Do the authors think this plays a role within their cohort? Please discuss.
25. Line 320: What does CCI stand for?
26. Line 352: Could you please provide data about the origin of the patients in the results section?
27. Ll 366-369: Please rephrase.
Author Response
Abstracts
Comments 1: Line 31-32: 1625 of 6959 patients is 23.4 %
Response 1:
Thank you so much for pointing this out. We have, accordingly, corrected the number in the text to match the number in Figure 1. We corrected 1625 (23.2%) to 1601 (23.01%) at line 32 and line 162, and corrected 5367 (76.8%) to 5358 (76.99%) at line 162.
Comments 2: Line 34: What do you mean with „more diabetes mellitus“? Patients who suffered from DM or patients who suffered more severely from DM?
Response 2:
Thanks for your reminder. To express it more clearly, we have revised „more diabetes mellitus“ to „more patients with diabetes mellitus“ at line 34.
Introduction:
Comments 3: Line 45: Please delete cavity
Response 3: We revised „oral cavity, sinus cavities” to „oral and sinus cavities” at line 46.
Comments 4: Line 57: Do you mean „local recurrence“ with „local failure“?
Response 4: At line 58, „Local failure” means the same as „local recurrence.” In Reference 5, the author expressed „local recurrence without distant metastasis" by „loco-regional failure (LRF)", so we revised to „loco-regional failure”.
Comments 5: Line 61: Please write „difficulties“
Response 5: Thanks for your reminder. We have revised „difficulty“ to „difficulties“ at line 62.
Material and Methods:
Comments 6: All numbers that are mentioned should be mentioned in the results section. The explanations given could remain in the Material and Methods section.
Response 6:
Agree. We revised text of line 90-91 to „Using the Chang Gung Research Database (CGRD), we searched non-metastatic head and neck cancer patients who underwent CCRT at four Chang Gung Memorial Hospitals in Taiwan.” Hence, we revised the first paragraph of “3-1 patient characteristics” to „Using the CGRD, we identified 9,729 non-metastatic HNSCC patients who underwent CCRT between January 2007 and December 2019. After excluding 631 patients based on exclusion criteria and 2,139 patients with NPC, we had 6,959 HNSCC patients remaining. Among these, 1,601 (23.01%) were identified as having pneumonia, while 5,358 (76.99%) did not experience any episodes of pneumonia,” at line 159-163.
Comments 7: Line 133: Please write „independent“^
Response 7: Thanks for your reminder. We have revised „Independent“to „independent“ at line 135.
Results:
Comments 8: Line 157: 23,4% see co Comments mment above.
Response 8: Just as response 1, we have corrected the number in the text to match the number in Figure 1. We corrected „1625 (23.2%)”to „1601 (23.01%)” at line 32 and line 162.
Comments 9: Line 158: 5367 of 6959 patients are 77.1%, not 76.8%. In figure 1 it is written that 5358 patients did not have a penumonia. Which number is correct?
Response 9: As response 1, we have corrected the number in the text to match the number in Figure 1. We corrected „5367 (76.8%)”to „5358 (76.99%)” at line 162.
Comments 10: Ll: 161-162: Please add the numbers of patients to the given percentages.
Response 10: We agree your comments. We added the numbers of patients and revised the line 165-167: “The primary tumor sites were distributed as follows: 3,273 cases (47.0%) in the oral cavity, 1,790 cases (25.7%) in the oropharynx, 1,489 cases (21.4%) in the hypopharynx, and 407 cases (5.8%) in the larynx.”
Comments 11: Table 1:
- Your cohort consists of only 5.9% females. It is known that males suffer more often from HNSCC. However, do you have any explanations for this major gender imbalance?
- Do you also collected pack years?
- Please write „Nasogastric“
- Please explain what is written in the table: data is shown in number of patients and percentages (you might delete „%“ in each cell), median with minimum and maximum or mean with SD
- Radiotherapy delay of more than 2 weeks?
- A legend with the main results of this table and explanation of the given numbers is missing. Please add.
Response 11:
- Thanks for your reminder. We added a sentence to explain this question at line 325-328 (and delete the sentence at line 360-363): „The markedly higher incidence of HNSCC in males compared to females in Taiwan is primarily attributed to the substantially greater prevalence of alcohol abuse , tobacco smoking, and betel nut chewing among men. And these risk factors also contribute to the poorer prognosis observed in male patients.”
- We apologize for not including the pack years data in our statistics, as the information in our database is incomplete.
- We agree your comment. We have revised „Nasgastric tube“ to „Nasogastric tube“.
- We agree your comment. We have deleted „%” in each cell and added legend in each table. The age is presented as the mean plus one standard deviation. Duration of chemotherapy (days) is presented as the median value with the minimum and maximum value. Duration of radiotherapy (days), albumin, and body weight are presented as the median values with one standard deviation values. The interval between the diagnosis of HNSCC and the initiation of CCRT (days) is presented as the median value with the difference between the maximum and minimum values. We have added the above description to the legends of the tables.
- Thank you for pointing that out. We revised „Radiotherapy delay of 2 weeks” to „Radiotherapy delay of at least 2 weeks” at table 1, 2, 3, and 4.
- We agree your comment.. We have added legends in each table to explain the given numbers and main results.
Comments 12: Ll 181 and 337l: Delay of at least 2 weeks?
Response 12: Thank you for pointing that out. We revised „delay of 2 weeks “ to „delay of at least 2 weeks” at line 208, revised „a two-week delay of radiotherapy” to „a delay of at least two weeks in radiotherapy” at line 381. We also revised „Radiotherapy delay of 2 weeks” to „Radiotherapy delay of at least 2 weeks” at table 1, 2, 3, and 4.
Comments 13: Line 182: A full stop is missing.
Response 13: Thank you for pointing that out. We have added the missing full stop at line 209.
Comments 14: Ll 175-182: This sentence is pretty long and hard to read.
Response 14: Thank you for pointing that out. We have rewritten the sentence and divided it into four sentences at line 200-209.
Comments 15: Line 185: Please write curve
Response 15: Thanks for your reminder. We have revised „cure“ to „curve“ at line 212.
Comments 16: Lines 200 and 217: What do you mean with „more alcohol drinking“/ „more alcohol consumption“? In table 1 and 2 you classified „never“ „former or current“
Response 16: Thanks for your reminder. Regarding „more alcohol consumption,” we have deleted „more” and retained only „alcohol consumption” at line 226 and 246, since it is already an independent prognostic factor in HNSCC patients undergoing CCRT.
Comments 17: Table 2-4: Legends are missing
Response 17: Thanks for your reminder. We have added legends in each table to explain the given numbers and main results.
Table 2. Univariate and multivariate analysis for Cox proportional hazard model of prognostic factors in HNSCC patients with CCRT. The categories that remain statistically significant after univariate and multivariate analysis include age, gender, pneumonia, alcohol consumption, N-classification 2 and 3, stage 4, diabetes mellitus, gastrostomy, nasogastric tube, duration of chemotherapy, radiotherapy delay of at least two weeks, and primary tumor site in the larynx. Abbreviations: HNSCC, Squamous cell carcinoma of head and neck; CCRT, Concurrent chemoradiotherapy; HR, Hazard ratio; C.I., Confidence interval; AJCC, American Joint Committee on Cancer.
Table 3. Univariate and multivariate analysis for Logistic regression of risk factors for pneumonia in HNSCC patients with CCRT (No matching). The categories that remain statistically significant after univariate and multivariate analysis include age, gender, alcohol consumption, N-classification 2 and 3, diabetes mellitus, gastrostomy, nasogastric tube, duration of chemotherapy, radiotherapy delay of at least two weeks, and primary tumor site in the hypopharynx. Abbreviations: HNSCC, Squamous cell carcinoma of head and neck; CCRT, Concurrent chemoradiotherapy; OR, Odds ratio; C.I., Confidence interval; AJCC, American Joint Committee on Cancer.
Table 4. After Greedy propensity score matching 1:2, univariate and multivariate analysis with Logistic regression of risk factors for pneumonia in HNSCC patients with CCRT. The categories that remain statistically significant after univariate and multivariate analysis include alcohol consumption, diabetes mellitus, gastrostomy, nasogastric tube, duration of chemotherapy, and radiotherapy delay of at least two weeks. Abbreviations: HNSCC, Squamous cell carcinoma of head and neck; CCRT, Concurrent chemoradiotherapy; OR, Odds ratio; C.I., Confidence interval.
Comments 18: Ll 214-236: This passage is hard to read and numbers are not the exactly the same like in table 3. Please correct the numbers in the table and delete them in the text as they are written in the table.
Response 18: Thanks for your reminder. We have reviewed all the numbers in the tables to ensure their accuracy, and we have streamlined the presentation of numbers in lines 225-231, 246-252, and 265-268 to improve readability.
Comments 19: Ll 232-238: Please delete the numbers in the text.
Response 19: As response 18, we have streamlined the presentation of numbers in lines 225-231, 246-252, and 265-268 to improve readability.
Discussion:
Comments 20: Line 258: What do you mean with „limited“?
Response 20: Thanks for your reminder. We have revised the whole sentence to “The limitations of these studies lie in the small number of patients enrolled” at line 290-291.
Comments 21: Ll 272-274: a verb is missing
Response 21: Thanks for your reminder. We added a „was“at the sentence: „Besides, in their study, there „was“no differences……group.” at line 308.
Comments 22: Line 275: Please write „was“.
Response 22: We have revised „is“ to „was“ at line 309.
Comments 23: Ll 276-282: Do the authors have any idea how having suffered from pneumonia around the time of diagnosis could influence the 10-year survival rates?
Response 23: As your suggestion, we added following sentences at line 386-396: “As described above in the article, developing pneumonia around the time of diagnosis is associated with a longer duration of chemotherapy and a higher rate of radiotherapy delays. This may be due to treatment interruptions caused by pneumonia, which could reduce treatment efficacy and lead to lower overall survival rates over long-term follow-up. Additionally, it may also be due to patients having more comorbidities or complications from the HNSCC itself, resulting in weakened immunity (such as diabetes mellitus) or anatomical abnormalities (caused by HNSCC itself, or by tracheostomy or nasogastric tube placement), making them more susceptible to pneumonia. Consequently, the lower long-term survival rate among pneumonia patients might be because those who develop pneumonia tend to have a poorer prognosis due to the underlying severity of their disease.”
Comments 24: Ll 304-319: In general the aging immune system loses the ability to protect against infections and cancer. Do the authors think this plays a role within their cohort? Please discuss.
Response 24: We added following sentences at line 347-352: “Evidence also suggests that HNSCC with higher cytotoxic CD8+ T cell infiltration have better prognosis, but the infiltration of cytotoxic CD8+ T cells decreases with aging. Bronchoscopic lavage examinations have also shown that CD8+ T cell infiltration in the lungs decreases with aging. This evidence indicates that immunosenescence may contribute to the increased incidence of HNSCC and the higher risk of lung infections with advancing aging.”
Reference:
Elmusrati, A., J. Wang, and C.Y. Wang, Tumor microenvironment and immune evasion in head and neck squamous cell carcinoma. Int J Oral Sci, 2021. 13(1): p. 24.
Meyer, K.C., The role of immunity and inflammation in lung senescence and susceptibility to infection in the elderly. Semin Respir Crit Care Med, 2010. 31(5): p. 561-74.
Comments 25: Line 320: What does CCI stand for?
Response 25: Charlson Comorbidity Index (CCI) is used to predict 10-year survival in patients with multiple comorbidities. We added “Charlson comorbidity index (CCI)” at line 364.
Comments 26: Line 352: Could you please provide data about the origin of the patients in the results section?
Response 26: Thank you for your reminder. We have included all the patient characteristics data in Table 1. However, our data does not specifically include racial or location statistics. Our hospital primarily treats local patients, who are almost all Taiwanese, a group of the Asian population.
Comments 27: Ll 366-369: Please rephrase.
Response 27: Thanks for your important comment. We revised line 421-424 to: “HNSCC patients undergoing CCRT are at an elevated risk of pneumonia and mortality due to factors such as alcohol consumption, history of diabetes mellitus, gastrostomy or nasogastric tube placement, prolonged chemotherapy duration, or radiotherapy delay of at least 2 weeks.”
Round 2
Reviewer 2 Report
Comments and Suggestions for Authors
Thank you very much for the reply. I think now the article improved a lot. Just one little mistake:
Line 58: Please correct and write "loco regional"